# Flexohand: A Hybrid Exoskeleton-Based Novel Hand Rehabilitation Device

**DOI:** 10.3390/mi12111274

**Published:** 2021-10-20

**Authors:** Tanvir Ahmed, Md Assad-Uz-Zaman, Md Rasedul Islam, Drew Gottheardt, Erin McGonigle, Brahim Brahmi, Mohammad Habibur Rahman

**Affiliations:** 1Biomedical/Mechanical Engineering Department, University of Wisconsin-Milwaukee, Milwaukee, WI 53211, USA; assaduz2@uwm.edu (M.A.-U.-Z.); rahmanmh@uwm.edu (M.H.R.); 2Richard J. Resch School of Engineering, University of Wisconsin-Green Bay, Green Bay, WI 54311, USA; islamm@uwgb.edu; 3Biorobotics Lab, University of Wisconsin-Milwaukee, Milwaukee, WI 53211, USA; dgottheardt@gmail.com; 4Physical Medicine & Rehabilitation Department, Medical College of Wisconsin (MCW), Wauwatosa, WI 53226, USA; emcgonigle@mcw.edu; 5Department of Electrical and Computer Engineering, Miami University, Oxford, OH 45056, USA; brahmib@miamioh.edu

**Keywords:** hand rehabilitation, isolated joints, isolated digits, fingers, thumb, compliant mechanism, composite motion

## Abstract

Home-based hand rehabilitation has excellent potential as it may reduce patient dropouts due to travel, transportation, and insurance constraints. Being able to perform exercises precisely, accurately, and in a repetitive manner, robot-aided portable devices have gained much traction these days in hand rehabilitation. However, existing devices fall short in allowing some key natural movements, which are crucial to achieving full potential motion in performing activities of daily living. Firstly, existing exoskeleton type devices often restrict or suffer from uncontrolled wrist and forearm movement during finger exercises due to their setup of actuation and transmission mechanism. Secondly, they restrict passive metacarpophalangeal (MCP) abduction–adduction during MCP flexion–extension motion. Lastly, though a few of them can provide isolated finger ROM, none of them can offer isolated joint motion as per therapeutic need. All these natural movements are crucial for effective robot-aided finger rehabilitation. To bridge these gaps, in this research, a novel lightweight robotic device, namely “Flexohand”, has been developed for hand rehabilitation. A novel compliant mechanism has been developed and included in Flexohand to compensate for the passive movement of MCP abduction–adduction. The isolated and composite digit joint flexion–extension has been achieved by integrating a combination of sliding locks for IP joints and a wire locking system for finger MCP joints. Besides, the intuitive design of Flexohand inherently allows wrist joint movement during hand digit exercises. Experiments of passive exercises involving isolated joint motion, composite joint motions of individual fingers, and isolated joint motion of multiple fingers have been conducted to validate the functionality of the developed device. The experimental results show that Flexohand addresses the limitations of existing robot-aided hand rehabilitation devices.

## 1. Introduction

Stroke, trauma, sports injuries, occupational injuries, spinal cord injuries, and orthopedic injuries are common prevalent occurrences in human life, often resulting in hand and finger impairment. The human hand is the most used external part of the human body for activities of daily living (ADL) [1,2]. A person’s life can be severely impacted by limitations of motion or even a tiny scar in their body [3]; the impairment of a hand causes a significant deficit in the performance of everyday tasks. Stroke reduces mobility in more than half of stroke survivors aged 65 and over [4]. In the United States alone, more than 610,000 people suffer strokes annually. Among stroke survivors, 50% are chronically disabled due to its high morbidity rate [5,6,7]. For most of these cases, partial or total loss of hand motor ability is observed. Functional recovery of the impaired upper limb is vital to regain independence and improve the quality of life. Rehabilitation programs are the primary method to promote active recovery in stroke and trauma survivors [8]. Conventional therapeutic approaches are well-established methods, but unfortunately, there is a constant shortage of trained clinicians/therapists to treat patients requiring long-term therapeutic intervention. Moreover, finding qualified therapists in underdeveloped countries can be even more of a challenge.

Rehabilitative therapy is a large piece of the puzzle in regaining independence among those suffering from upper extremity limitations. Passive exercise is the first step towards the recovery of hand function after stroke and trauma. Our customer discovery [9] with individuals with upper/lower limb dysfunctions showed that (i) patients often cannot comprehend or perform exercises at home correctly; (ii) time commitment required by the patients/family caregivers in the rehabilitation program, and their financial constraints often cause the patients to withdraw from the rehabilitation program, posing a significant socioeconomic burden; and (iii) travel, transportation, and limited insurance coverage constrain the patients from having adequate rehabilitation and is the primary reason for patients’ dropout from a rehabilitation program. These dropped-out individuals, as a result, depend on caregivers/family caregivers for their essential ADL care. Therefore, there is a pressing need to innovate alternate rehabilitation treatment schemes that address the shortcomings of the current rehabilitation therapy delivery practice.

As an alternative or supplement to the conventional method of treatments, robotics technologies have emerged [10,11,12,13,14,15] to augment the recovery process and facilitate the restoration of hand function. One form of treatment involves continuous passive motion (CPM) of the digits to help restore motion. CPM involves passively ranging the fingers with the use of external forces. The more significant benefit is achieved when performed for relatively longer sessions in the neighborhood of 45 min or so. Thus, this form of treatment is generally reserved for independent administration at the patient’s home so as not to monopolize the time spent in therapy with clinicians. As an adjunct modality, CPM can have positive effects on regaining digit motion more rapidly. The advancements in robotics further expands the usability of such modalities.

Stroke, trauma, or even finger surgery often result in stiffness of finger movement and restrict hand functions, impacting the affected person’s ability to perform activities of daily living. To regain mobility in the affected hand, therapists often prescribe nerve gliding in stroke survivors [1], who cannot actively make finger movements due to muscle fatigue. However, their fingers can be moved passively within the natural range of motion. Typical tendon gliding exercises are often a combination of knuckle bend, hook fist, and straight fist exercises where all hand digits move simultaneously. Finger trauma, such as fractures, usually requires isolated joint ROM exercises in addition to tendon gliding exercises [16]. Other examples of digit ROM exercises include, but are not limited to, full composite flexion of all the digits commonly known as imaginary ball squeeze or claw hand exercises. Independent flexion and extending of an isolated finger joint are often used as a therapeutic exercise. Depending on the patient’s specific condition, therapists design the rehabilitation therapy scheme to regain hand and digit function. Such versatile therapies can be translated in terms of digit joint-specific motions so that robot-aided therapy can be used as an effective mode of intervention. In Table 1, various hand therapies and associated digit joint motions have been summarized to determine the requirement of a robotic device for hand rehabilitation.

Robotic devices capable of providing continuous isolated and combined digit joint motions to all digits can expand the horizon of robot-aided hand rehabilitation both in a clinical setting and at home. Furthermore, ease of wearability and portability of the device can increase the efficacy of robot-aided home-based rehabilitation. An important factor that cannot be ignored in dealing with digit ROM limitations is that many of the digit tendons travel across the wrist; changing wrist and forearm positions can alter the dynamics of how the tendons work. Some of these motions include wrist flexion/extension, radial and ulnar deviation, and forearm pronation/supination. Additionally, wrist and forearm positions are often required or encouraged to achieve full potential motion during patients’ daily life activities. Therefore, restricting these motions of the wrist can hinder the potential of robot-aided therapy. Another key issue in the continuous passive motion of finger using devices is that passive MCP abduction–adduction motion naturally occurs during MCP flexion–extension. These passive motions should be considered while developing such devices for hand rehabilitation.

Robotic device-aided upper limb rehabilitation has been very popular for reducing the burden of going through therapy [17,18,19,20,21,22,23,24]. This mode of rehabilitation has shown great promise among therapeutic interventions. Robotic rehabilitation devices are mainly of two types: end-effector/endpoint type [11,25,26,27], and exoskeleton type [28,29,30,31]. The end-effector type devices for hand rehabilitation must remain stationary, and the patient is required to place the affected hand onto the device to receive the treatment. End-effector/endpoint devices for hand rehabilitation [32,33,34] are mechanisms that act on the distal tip of fingers propagating motion to DIP, PIP, and MCP joints. These devices can accommodate a variety of hand sizes, but isolated finger movement cannot be achieved effectively. Due to their easily manufacturable design, quite a few end-effector type devices have become commercially available on the market, commonly known as continuous passive motion (CPM) devices. Currently, CPM devices such as Waveflex CPM [35] and Kinetec Maesta [36] are used in clinical settings and at home and are often covered by Medicare or other health insurance policies. However, these products cannot provide isolated flexion and extension movement to finger joints. There are a few commercially available devices such as Reha-Digit [37] and Amadeo [38] which can provide isolated finger ROM but cannot offer isolated joint motion. Vinesh et al. [39] developed a non-actuated sensored hand glove integrated with a computer game (Flappy Bird) to engage patients playing a game where the subject’s single/multiple fingers are involved, representing fine motor skill occupational therapeutic exercises. There are also some non-actuated peripherals for hand rehabilitation available on the market, such as the SAEBO Glove [40] and MusicGlove [41], which functions to strengthen the finger muscles. Still, these cannot provide any passive movement to the patients’ hand, which is paramount towards recovery from hemiplegia due to stroke.

Due to the limitations of end-effector type devices, over the past few years, researchers have been leaning towards exoskeleton type robotic devices for hand rehabilitation [42]. Exoskeleton-based design approaches are more suitable for generating isolated finger joint motions and digit movements but can become quite complex due to hand morphology. The bones of the human hand can be quite small while having 27 joints and associated degrees of freedom (DoF). Even when only the flexion–extension motion of DIP, PIP, and MCP joints of fingers and IP and MCP joints are considered, 14 DoF need to be accounted for. A wearable exoskeletal rehab device provides motion to digit joints by maintaining virtual joint axes during motion or aligning joint axes of the structural parts with the digit joint axes. Gonzalez et al. developed a novel virtual joint-based exoskeletal device, ExoK’ab [43], capable of providing isolated motion to digits and fingers’ PIP and MCP joints and the thumb’s MCP joint. Their device utilized a combination of worm-geared motors and a telescopic mechanism mounted on the forearm-supported base. The user’s hand is attached to the device using Velcro straps at the middle and proximal phalanges during exercises. The ExoK’ab adds 731 g of wearable weight to the user’s hand, and the base structure restricts any wrist motion during hand exercises. Virtual joint-based design [43,44,45,46,47] requires extensive integrated parts to make sure the virtual axis of rotation of the exoskeleton matches the human hand during flexion–extension motion. Soft robotic devices based on artificial muscles [48] or tendons [49,50] have shown great promise in designing simpler mechanisms capable of producing digit motions in the user’s hand. These devices produce the external forces required to achieve ROM without utilizing solid structural parts, alleviating the necessity for maintaining a virtual joint axis through an external mechanism. It should be noted that the pneumatic muscle-based SIFREHAB marketed by SIFSOF, US [51] is commercially available in the market. However, this device lacks a provision for practical tendon glide exercises that require isolated joint movements. In addition, artificial muscles actuated through pressurized elements may leak and reduce system reliability over time unless explicit maintenance is carried out. Electrically actuated tendon-driven soft robotic devices for hand rehabilitation, such as those from Bernocchi et al. [50] and Chen et al. [49], have the same issue of not being able to provide isolated joint movements. Exoskeleton type rehab devices with aligned joint-based mechanisms can provide isolated digit joint motion while having fewer structural parts than virtual joint axis-based mechanisms. But this approach requires space at the sides of the finger for positioning structural elements. This approach is suitable for the index finger and thumb where there is enough space to demonstrate the workability of the designed device, but applying the exact mechanism for the middle, ring, and small fingers poses a problem due to space restriction between index-middle, middle-ring, and ring-small fingers. This issue is further compounded by the fact that all four fingers come together to achieve a full range of motion, namely a full fist, due to passive MCP adduction reducing the inter-finger space even more. Many researchers [52,53] have demonstrated good motion and control of the index finger and thumb. Still, we have yet to see the application of those novel designs to rehabilitate middle, ring, and small fingers. Moshaii et al. [54] have shown a scheme for isolated phalange motion in addition to isolated digit motion. Their design is such that the user’s hand is fixed on a stationary platform that restricts wrist motions during finger motion therapy.

In this research, a robotic device, namely “Flexohand,” has been developed to fulfill the therapeutic needs (see Table 1) for hand rehabilitation. Flexohand addresses the limitations of current solutions comprising both research prototypes and commercial solutions based on the following criteria:Isolated and combined digit motion of fingers: (index, middle, ring, small) and thumb;Isolated and combined digit joints flexion–extension (fingers: DIP, PIP, MCP; thumb: IP, MCP);The device should not restrict wrist motion;The device should accommodate natural motion during finger flexion–extension by compensating MCP abduction–adduction motion;Easy donning and doffing;Lower added weight burden to the user’s hand.

The main contribution of this research is the development and incorporation of a novel compliant mechanism for passive compensation of MCP abduction–adduction during fingers’ MCP flexion and extension exercises. In addition, a combination of sliding locks for IP joints and a wire locking system for finger MCP joints have been integrated for achieving isolated and composite digit joint flexion–extension. Finally, a tendon transmission system has been designed to reduce the wearable weight of the device. Moreover, this system allows wrist joint motions during hand digit exercises. The rest of the paper is structured as follows: Section 2 presents a detailed description of the Flexohand. The kinematic modeling of Flexohand and the relationship between joint angles and actuator rotation have been presented in Section 3. Section 4 describes the donning and doffing method of the device. The experimental evaluation and discussion are summarized in Section 5. Finally, the paper ends with the conclusions presented in Section 6.

## 2. Anatomically Inspired Design

Anatomically, human fingers are classified into two types: the thumb and the other four fingers. The human thumb consists of three joints: the interphalangeal (IP) joint, metacarpophalangeal (MCP) joint, and the trapeziometacarpal (TMC) joint (see Figure 1A). Anatomically, index, middle, ring, and small fingers differ from the thumb. Each finger is composed of three joints: the distal interphalangeal (DIP) joint, the proximal interphalangeal (PIP) joint, and the metacarpophalangeal (MCP) joint (see Figure 1B). The range of motion (ROM) of the digits is achieved by the contraction of muscles which generate the necessary force for movement, and tendons transmit muscular forces to the joints, which induce flexion and extension of the fingers. Muscles are connected to tendons, which are connected to bones at their insertion points and the muscles’ origin point. Annular ligaments or pulleys serve to ensure the tendons stay in the correct path or position in the fingers and amplify the pulling force of finger flexion (Figure 2). The primary finger motions are flexion, extension, abduction, abduction, and rotational movements. For designing Flexohand, only flexion and extension of the finger joints were considered. Two types of muscles and tendons are responsible for such motion. Extensor digitorum muscles and extensor tendons are responsible for the extension motion, and flexor digitorum muscles and flexor tendons are responsible for the flexion motion. Each finger of the hand consists of various bones and joints and can be considered a robotic manipulator. Where muscles actuate revolute joints, power is transmitted by tendons, and the pulleys guide the tendons.

In our design, we leveraged the knowledge of human anatomy by developing a compliant exoskeleton type device. The index, middle, ring, and small fingers exoskeleton is composed of a distal phalange shell (DPS), middle phalange shell (MPS), and proximal phalange shell (PPS), and the thumb exoskeleton is composed of a DPS and PPS. Figure 3 illustrates the associated shells for housing finger phalanges. The DIP and PIP joints of the exoskeleton are aligned with the axis of rotation of finger joints.

The flexor sheathing of the DPS, PPS, and MPS segments are mechanical versions of pulleys in the human hand (from Figure 4). In the DPS, PPS, and MPS, the angled section serves as a hardware limit that eliminates the possibility of moving the mechanism to a position beyond the human’s anatomical ROM, denoted as “Flexion Limits” in Figure 3.

The open type shells are designed to accommodate the finger phalanges in such a way that, while donning the device, the finger slides into the associated exoskeleton shells and remain housed in the shells during flexion–extension movement of the finger phalanges by the extended part of the DPS, PPS, and MPS (Figure 4). The extended part of the shells encompasses the palmar region of the finger exterior between the interphalangeal digit creases. This approach reduces the need for adding Velcro straps or other methods of keeping the exoskeleton connected with the fingers during rehabilitative exercises. These extended parts are also used as a sheath to pass the gliding flexor wire.

Figure 5 shows grooves in the DPS, PPS, and MPS for accommodating the accumulated skin of DIP and PIP knuckles during the extension of these joints. These parts are pivoted against each other with a simple extruded part and hollow ring type structure. This pivotal joint of the exoskeletal structure negates any longitudinal force exerted on the finger digits by the device during flexion–extension motion of fingers.

### 2.1. Compliant Mechanism

Human fingers have varying gaps between adjacent fingers during flexion–extension, which conforms to natural hand motion. This gap is lowest when making a fist with a hand. The extruded portions of the exoskeletal shells at the DIP and PIP joint occupy a 7 mm space between index-middle, middle-ring, and ring-small fingers. During the flexion motion of fingers, the exoskeleton of each finger comes together due to the passive adduction of MCP joints. This causes mechanical interference between two adjacent exoskeleton modules. Therefore, we have designed a novel MCP-compliant mechanism. The MPS of each finger is connected to the respective MCP-compliant module via a frictional sliding lock (Figure 6). For a specific user, the relative position of the associated MPS and MCP-compliant module is adjusted by external force for their first time donning the device. The device retains this position for future usage via interbody friction between the MPS and MCP-compliant modules. This adjustability serves three key purposes: (i) the DIP and PIP joints of the index, middle, ring, and small finger and associated exoskeletal segments can be aligned properly; (ii) individual fingers’ exoskeletons do not collide during isolated or multi-finger movements; (iii) minimal resistive force is generated by the MCP-compliant mechanism during MCP flexion–extension while maintaining the hand’s natural motion during the movements; and (iv) passive compliance of MCP abduction–adduction of the index, middle, ring, and small finger during flexion–extension exercises.

The four MCP-compliant modules are connected via a general-purpose elastic cord. One elastic cord of Ø2.5 mm diameter is routed through the compliant modules that pass through three holes in each module and is locked at the outer side of the index and small finger’s MCP modules. The use of a single elastic cord allows uniform force distribution through all four fingers’ MCP-compliant modules during passive compliance of MCP abduction–adduction during MCP flexion–extension motion. The frictional sliding lock in the MCP-compliant modules’ slots are angled so that all four fingers are spread out during fingers’ extension. This reduces interference due to friction between two adjacent exoskeletal shells. The orientation of the MCP-compliant mechanism can be seen in Figure 7a,b. The elastic cord is tensioned so that during the finger extension motion, unless externally actuated, the MCP compliant modules pull towards each other, keeping the fingers apart (Figure 7a). During MCP flexion of fingers, the changing gap between adjacent finger exoskeletons is passively accommodated by the MCP-compliant mechanism (Figure 7b), ensuring reduced collision during the motion. Figure 7c shows the middle MCP-compliant module with an oriented frictional sliding lock for the connecting MPS.

### 2.2. Transmission and Actuation Mechanism

We implemented sets of flexor and extensor tendon wires comparable to extensor digitorum muscles and flexor digitorum profundus muscles in the hand. The flexion wire is routed through the flexor wire sheathings (Figure 8) at the palmar side of the DPS, PPS, and MPS through the palm module using a Bowden tube towards the motor assembly. Similarly, the flexor wire is routed through the flexor wire sheath at the dorsal side of the DPS, PPS, and MPS and then passed through the back palm module towards the motor assembly. The back palm module is worn at the dorsal side of the hand, and the palm module is worn at the palmar side of the hand.

The end of the flexor, extensor, and MCP lock wire is connected to the V-grooved disk directly related to the motor hub. For each finger (index, middle, ring, and small), a set of three motors (Lewansoul LX-16A [55]) was used (see Figure 9). Two motors, namely the flexor motor and extensor motor, are responsible for providing flexion and extension motion, and the third motor is used for restricting the movement of the MCP joint. For the thumb, we used two motors for flexion and extension. In total, this prototype of Flexohand uses 14 actuators which are mounted on a motor assembly board. The flexor, extensor, and MCP lock wires are connected to motors so that when the motor rotates counterclockwise (CCW), the wire is pulled respective to the Bowden tube generating tension. In cable-driven transmission systems, there is a very high possibility of cable slag and self-winding. To solve this issue, both the flexor and extensor motors work together during flexion and extension. For flexion motion, the flexor wire is pulled by the flexor motor’s CCW rotation, and the extensor motor rotates clockwise (CW) at a higher velocity to prevent slack in the wire.

The MCP lock wire, when pulled by the MCP lock motor, restricts the motion of the MCP joint. The MCP slider lock can slide and be positioned between two adjacent MCP-compliant modules to make the finger exoskeleton modules rigid to improve donning and doffing. For the thumb, we only considered IP and MCP flexion–extension. Therefore, the motion of the carpometacarpal (CMC) joint was limited by a thumb CMC brace. For the thumb exoskeleton, a similar flexor and extensor wire is routed through the Bowden tube connected to the CMC brace.

### 2.3. Isolated Digit and Digital Joint Motion

In this hand rehabilitation device, any desired isolated finger motion or isolated joint motion can be achieved by restricting the movements of the other unintended finger joints by configuring the motors’ position. To achieve isolated finger motion, the motors associated with other fingers are actively kept at zero position while the desired finger’s associated motors rotate according to the positional command. The isolated joint motion is achieved by introducing a simple slider lock for DIP and PIP joints and a cable-based lock mechanism for the MCP joint. When DIP motion is to be restricted, the associated DIP slider lock (Figure 3) is pushed between DPS and PPS. Similarly, to lock PIP motion, the PIP slider lock is moved between PPS and MPS. Removing the DIP/PIP slider lock frees that joint, allowing that joint’s motion. To restrict the motion of MCP joints, the MCP lock motor is kept fixed at its position while the MCP joint is at the extension position. For DIP and/or PIP joints’ flexion motion, the flexor motor pulls the flexor wire, the extensor motor releases the extensor wire, and the MCP joint is restricted, resulting in flexion–extension motion of either the DIP, PIP, or both joints based on the configuration. During flexion motion of the finger, the majority of the tension generated in the flexor wire by the flexor motor first acts on the MCP joint, causing MCP flexion. However, as the flexor wire is passed through the palm module, when the MCP joint is fully flexed, the flexor wire encounters a high frictional force that limits the movement of the DIP and PIP joints. Therefore, during DIP and/or PIP flexion motion, the MCP joint is locked until DIP and/or PIP flexion has been achieved. Afterward, the MCP motor releases the MCP lock wire simultaneously with the flexor motor’s flexor wire winding, thus allowing for the flexion of all the joints. For extension motion, the flexor motor and MCP lock motor release the associated tendon wires at a higher velocity while the extensor motor pulls the extensor wire at a comparatively lower velocity. Various isolated and composite finger joint motions and associated device configurations have been shown in Table 2.

### 2.4. Modelling of Structural Parts

Anthropomorphic references from the healthy adult (participant-A, age: 30 yrs., height: 64 in., weight: 152 lbs.) subject’s right hand were taken before designing the Flexohand. Exoskeletal shells, locks, and other modules were designed in Creo Parametric software, 6.0.2.0, PTC, Boston, MA, USA. It should be mentioned that, while designing such exoskeleton type devices for finger rehabilitation, it is essential that the device’s structural joint pivots and the hand digits’ rotation axes are aligned. The exoskeletal shells were designed to conform to this requirement with the use of data from the subject. The DIP and PIP/IP joint axes of the hand digits were aligned with the DPS-PPS and PPS-MPS pivotal joints. Fingers’ MCP joint axes of rotation are compensated by the compliant mechanism and, in the case of the thumb, the combination of a thumb brace and the orientation of tendon wires ensures that thumb MCP flexion–extension is achieved nominally. It is to be noted that the current prototype of Flexohand is specific to an individual, participant-A. For a different participant, associated anthropomorphic parameter values would need to be updated. The parametric design capability of Creo Parametric software enables us to input the updated parameter values into the CAD environment and achieve new models of exoskeletal shells and compliant mechanism parts which are specific to each participant. Afterward, the new parts with different dimensions can be 3D printed and assembled for usage.

## 3. Kinematic Analysis

Each finger combined with its associated exoskeleton can be described as a 4 DoF (2R-R-R) serial manipulator where MCP abduction–adduction, MCP flexion–extension, PIP flexion–extension, and DIP flexion–extension motions have been considered. In contrast, the thumb can be defined as a 2 DOF (R-R) serial manipulator considering IP flexion–extension and MCP flexion–extension motions. Figure 10 shows the link frame assignment for a finger and exoskeletal segments where *L*_1_ = the length of the proximal phalange, *L*_2_ = the length of the middle phalange, and *L*_3_ = distance between the DIP joint and fingertip. In Table 3, we have summarized the modified Denavit–Hartenberg (DH) parameters [56] associated with the developed kinematic model.

The transformation matrix for the kinematic model is expressed using Equation (1). Here, $\left[\begin{array}{c} P_{x}\\ P_{Y}\\ Pz \end{array}\right]$ defines the position of the fingertip respective to the corresponding MCP joint.
(1)Tf0=[12(cosσ3+σ2)−12(sinσ3+σ1)sinq1Px12(σ1−sinσ3)12(σ2−cosσ3)−cosq1Pyσ4σ60Pz0001]
where $P_{x},P_{y},P_{z},\;\sigma_{1},\;\sigma_{2},\sigma_{3},\sigma_{4},\sigma_{5}$, and $\sigma_{6}$ can be found in Appendix A.

In this device, the MCP abduction/adduction associated angle q_1_ adjusts passively using the compliant mechanism during MCP flexion/extension. Therefore, each finger exoskeleton can be defined as a simple 3 DoF (R-R-R) planar manipulator. This paper focuses on estimating digit joint angles by developing a relation between motor rotation and effective tendon wire length.

Angles corresponding to digit joints’ flexion–extension motion can be defined by relating joint angle values to the varying effective tendon wire length responsible for achieving the motion. Figure 11 shows the procedure for estimating isolated DIP joint flexion angles. To estimate flexion angle first, we drew a circle with radius OA = OB = r, where O denotes DIP joint’s center of rotation, A denotes the flexion wire exit point of DPS, B denotes the flexion wire entry point of PPS, OA is the distance between O and A, and OB is the distance between O and B.

Then, we constructed a straight line connecting A and B and finished the mathematical model formulation by drawing another straight line, OC ⊥ AB, passing through O. ACB is the effective flexor wire length during any intermediate positions during DIP joint motion (see Figure 11(ii)). For both DIP extension and intermediate positions, AC = AB and AB = AC + CB; thus, ΔOAB is an isosceles triangle in these cases. During DIP flexion (see Figure 11(iii)), the points A, C, and B coincide together; therefore, OA = OC = OB.

Using Figure 12, we derived the relation between the DIP flexion angle, δ = 2 β (left) and the flexor motor’s angular rotation, *θ* (right). The left figure corresponds to the DIP joint’s extended position (see Figure 11(i)), where the angle between DPS and PPS, δ = ∠AOB, is maximum. According to the shell design in the CAD environment, we found ∠AOB = δ = 80°, and therefore β = 40° and OC = 5.5 mm.

Knowing β allowed us to find α, as ∠OCA = ∠OCB = 90°, and then we calculated the radius of the constructed circle, *OA*, for the DIP extended configuration, using Equation (2) and cord length, and *ACB* using Equation (3):(2)$$OA=OB=c=\frac{a}{\sin\alpha }$$
(3)$$ACB=L=2b=2\;\sqrt{c^{2}-a^{2}}$$

Then, we found the varying length of line *ACB* = *L*′, corresponding to varying ∠AOB = δ’ for different DIP intermediate positions using Equations (4) and (5):(4)$$\alpha^{\prime }=90{}^{\circ}-\beta^{\prime };\;where\;\beta^{\prime }=\frac{\delta^{\prime }}{2}$$
(5)$$a^{\prime }=c\sin\alpha^{\prime }$$
(6)$$L^{\prime }=L-∆L=2b^{\prime }=2\sqrt{c^{2}-{\left(a^{\prime }\right)}^{2}}$$
where ∆*L* is the relative change in effective tendon wire length.

In Figure 12, ACBQP is the total flexor wire length where, during all positions of DIP flexion, the angle length of the BQ section remains constant. According to the flexor wire’s connection to the flexor motor (see Figure 8 and Figure 12), the relative decrement of effective tendon wire is achieved by rotating the flexor motor counterclockwise and thus varying *θ* = the motor’s angular position. The flexor wire end is connected to the V-grooved disk via wire lock at point P. Q is the virtual fixed point where the flexor wire always touches the V-groove due to tension. O’ denotes the center of rotation of the flexor motor, and R is the effective radius of the V-grooved disk. The flexor wire is connected such that when the DIP joint is fully extended, point P and Q coincide, resulting in *θ* = 0. As we increase *θ*, *S* increases, and due to the connection configuration, we can conclude the following:(7)S=∆L 
where, *S* = The arc length between P and Q.

We calculated *S* respective to *θ* using Equation (8):(8)S=Rθ

Using Equations (4)–(8), the relation between the flexor motor’s angular position, *θ*, and the DIP joint’s flexion angle, δ’, for isolated DIP joint flexion motion can be formulated as Equation (9):(9)$$\theta =\frac{1}{R}\left[2b-\frac{2a}{\sin\alpha }\sqrt{1-{\left(cos\frac{\delta^{\prime }}{2}\right)}^{2}}\right]$$

The same approach is used for determining the required associated motors’ rotation for isolated DIP extension, isolated PIP flexion–extension, and isolated MCP flexion and extension. Note that, with this approach during composite motion, DIP and PIP flexion–extension, the DIP and PIP angles cannot be individually computed, so in this case, we describe the flexion–extension motion by summing DIP and PIP joint angle values. Furthermore, with this approach, the computed MCP joint angles are expected to be less accurate due to nonuniform positions of the MPS tendon wire exit points and back palm and palm modules’ tendon wire entry points. During experimentation, the motor angular position values, *θ*, were obtained to generate the intermediate position of joint angles, δ’, from the corresponding Equation (9) for DIP, PIP, and MCP joints.

## 4. Donning and Doffing of the Device

Donning this rehabilitation device is akin to wearing a glove (Figure 13). First, the user needs to wear the thumb exoskeleton module mounted on the CMC brace by strapping Velcro straps around the palm. Secondly, the finger exoskeletons are locked together by sliding the MCP sliding locks between the adjacent fingers’ respective MCP lock modules. This creates a rigid structure for increased wearability. Then, the user slides their fingers (index, middle, ring, and small) into the respective finger exoskeletons. Afterward, the palm module and back palm module are strapped together around the hand using Velcro straps. The critical point here is to tighten the straps to the degree that the compression due to the strap does not restrict the motions of the tendons in hand. Finally, the elastic cords connected to the palm module and back-palm module are pulled around the wrist, so the modules are fixed to the hand. Afterward, the MCP sliding locks are slid back into the respective fingers’ MCP compliant module, allowing the MCP compliant mechanism to work freely during exercise. To take off the device, the steps mentioned above are done in reverse. Experimentally, we have found that it takes about one and half minutes to don and doff this device from the user’s hand.

## 5. Experimental Evaluation

### 5.1. Fabrication abd Experimental Setup

The modelled parts were printed using an SLA type 3D printer (printer: Elegoo MARS [57]; material: UV-curing photopolymer rapid resin [58]). Nylon-coated fishing wires were used as tendon wires and a commercially available Bowden tube (PTFE) was used as a tendon wire shell. This device has low wearable weight, meaning when the user wears the device, it adds around 280 g of weight to the user’s hand. The motor assembly can be placed on any stationary surface, which reduces the burden on the user’s hand. The schematic of the experimental setup of the device can be seen in Figure 14. The motors are controlled by using Arduino Sketch running on a personal computer. For different therapeutic exercises and ranges of motion of the hand digit joints for a user, the program input parameters can be modified by anyone with access to the motion program at the current stage of development. Note that, in the future, the device is intended to be used for personalized therapy where both the user and therapist can modify the motion program from a graphical user interface (GUI).

### 5.2. Actuation Calibration

This device prototype was tested on participant-A, and a variety of isolated finger, isolated finger joint, and composite joint motion exercises were performed. During the experiments, we found that the required angular position of the actuators (flexor, extensor, and MCP lock motors) to achieve deterministic joint flexion–extension angles varies from the angular position obtained by using Equation (9). These deviations are caused by the Bowden tube shell being considered a rigid shell during the formulation of Equation (9) with respect to joint angles. In contrast, in the experiments it was found that, during flexion–extension motions, the tension generated in the tendon wires causes the Bowden tube to deform, resulting in variance in the required effective tendon wire length. Therefore, the flexor, extensor, and MCP lock motors’ angular positions were determined empirically and calibrated using goniometers for experimentation. To calibrate joint angles, we sampled motor position combinations five times. Each time the joint angles were manually measured using a goniometer. Afterward, we average the angle values to generate a calibration chart associated with the different motions of each digit in the hand. We present the calibrated motor angle values and associated joint angles for fingers in Table 4. Note that human errors are to be expected as the measurements of joint angles are taken manually. Therefore, we define the calibrated joint angle values as a close approximation to actual digit joint angles. Note that the calibration data has been generated for participant-A, and it was a one-time process. For subsequent usage and experimentation with participant-A and the developed Flexohand, the calibrated data remains valid.

### 5.3. Experiments with Flexohand

The efficacy of this developed prototype was evaluated based on the device’s capability to generate various isolated and composite motions of digit joints. Using the calibration chart, positional commands are sent to motors to achieve respective joint motions. A selected few joint ROM exercises among all possible configurations are presented in Figure 15, Figure 16, Figure 17, Figure 18, Figure 19 and Figure 20, where (a) shows the initial position, (e) shows the final position of each exercise, and (b, c, d) show the intermediate positions between initial and final positions. DIP flexion of the middle finger has been shown in Figure 15, where the PIP joint is locked via a PIP lock, and the DIP lock is removed. Then, a positional command is passed to the motor control module for achieving DIP flexion motion. Figure 16 shows the PIP flexion of the index finger, where the PIP joint is unlocked by sliding the PIP lock towards the DIP joint and removing the DIP joint lock.

In the case of thumb MCP extension (see Figure 17), the IP joint is locked, and the thumb MCP joint angle is changed from 25° to 0°. Similarly, PIP flexion of the index, middle, ring, and small finger are shown in Figure 18. Figure 19 shows the simultaneous MCP flexion of all fingers and thumbs where all DIP and PIP/IP joints are locked. To achieve composite flexion of the DIP, PIP, and MCP joint, all interphalangeal locks (finger: DIP, PIP; thumb: IP) are removed. Then, DIP and PIP flexion is achieved in Step 1, and finally, MCP flexion is achieved in Step 2 (see Figure 20).

From the experiments conducted in this research, it was seen that the participant was able to receive various passive hand digit exercises by using the prototype of Flexohand. These passive exercise routines were generated to show the therapy providing capabilities of the developed mechanism. These routines comprise various isolated joint motions, composite joint motions of individual fingers, and isolated joint motions of multiple fingers. The series of snapshots taken from the video recorded during experimentation shows some of the therapy routines. The wrist joint is not restricted during the finger exercises as the tendon sheath does not travel across the wrist. With a combination of sliding locks and MCP locking mechanism, various isolated joint and composite joint motions are achieved with Flexohand.

## 6. Conclusions

In this research, a robotic device for hand rehabilitation, namely Flexohand, was developed. Flexohand incorporates multiple mechanisms for providing isolated digit joint motion of all fingers and the thumb. The prototype of Flexohand was built using low-cost 3D printers, printing materials, and actuators. The current prototype was used to provide various rehabilitative passive therapies to participant-A. The efficacy of the mechanism used for Flexohand will be evaluated further by improving fabrication processes and with the addition of better actuators.

## Figures and Tables

**Figure 1 micromachines-12-01274-f001:**
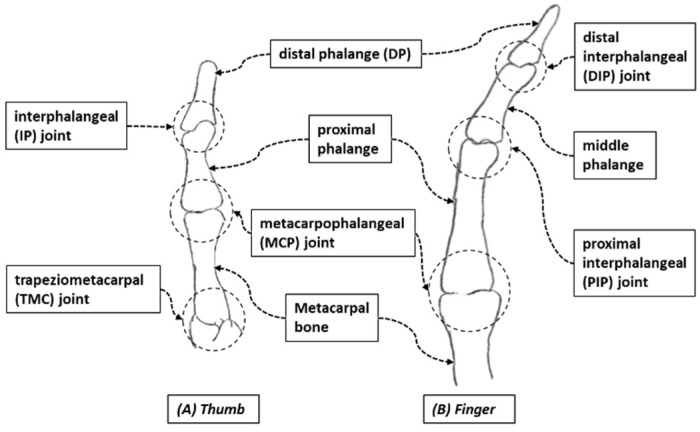
Structure of finger (**B**) and thumb (**A**).

**Figure 2 micromachines-12-01274-f002:**
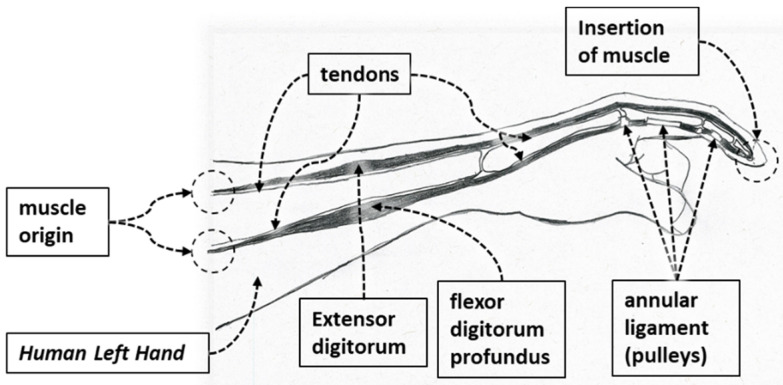
Working mechanism of fingers through muscles, tendons, and pulleys.

**Figure 3 micromachines-12-01274-f003:**
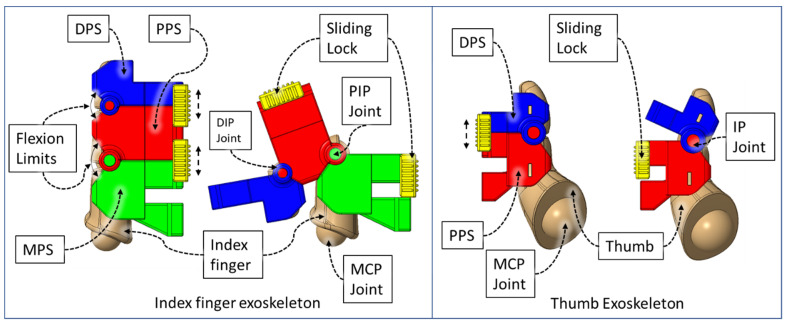
Developed finger and thumb exoskeletons.

**Figure 4 micromachines-12-01274-f004:**
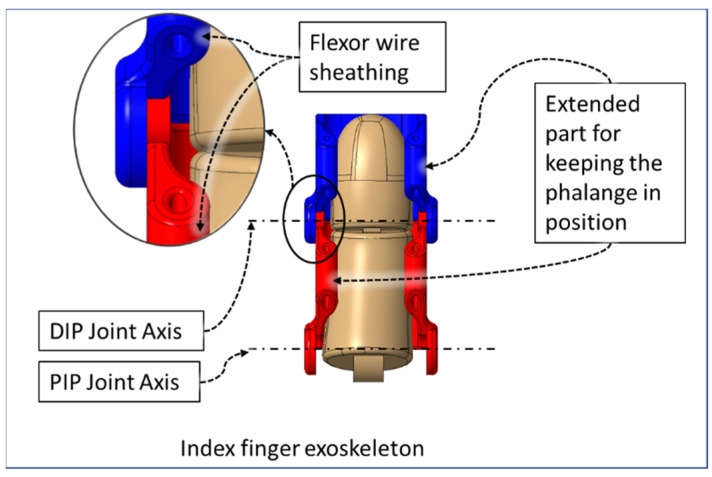
Palmar view of finger exoskeleton segments.

**Figure 5 micromachines-12-01274-f005:**
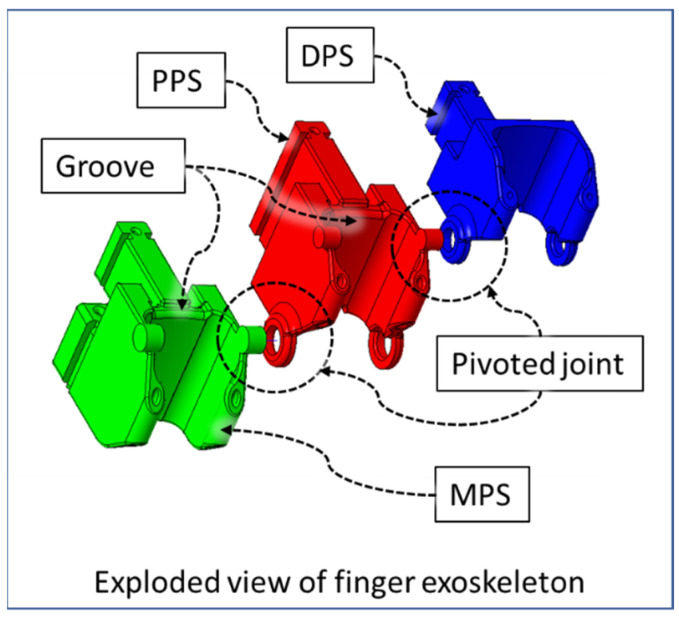
Exploded view of finger exoskeleton.

**Figure 6 micromachines-12-01274-f006:**
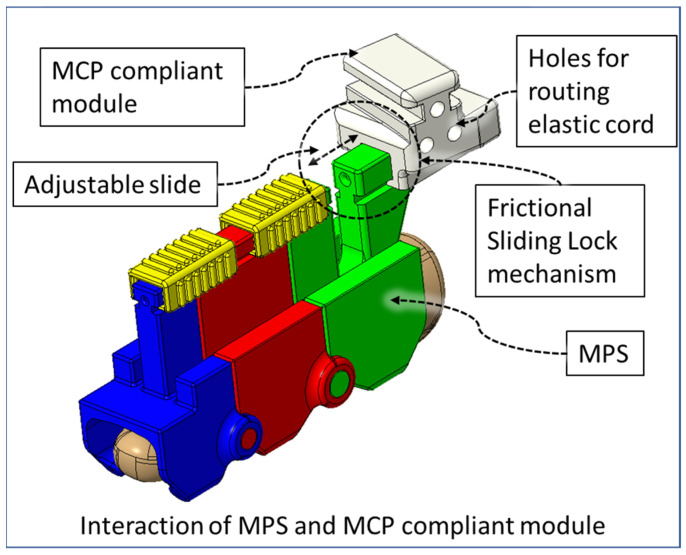
Frictional sliding lock between MPS and MCP compliant module.

**Figure 7 micromachines-12-01274-f007:**
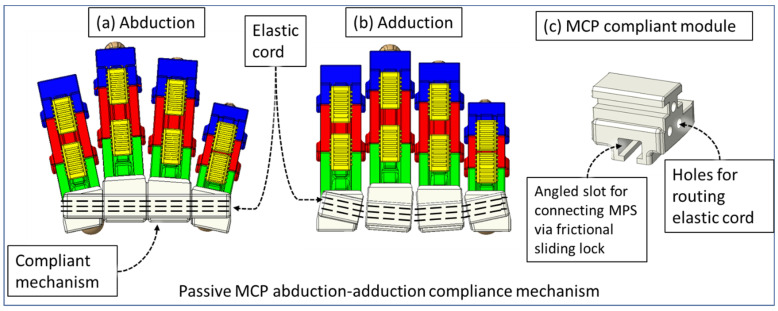
Novel compliant mechanism.

**Figure 8 micromachines-12-01274-f008:**
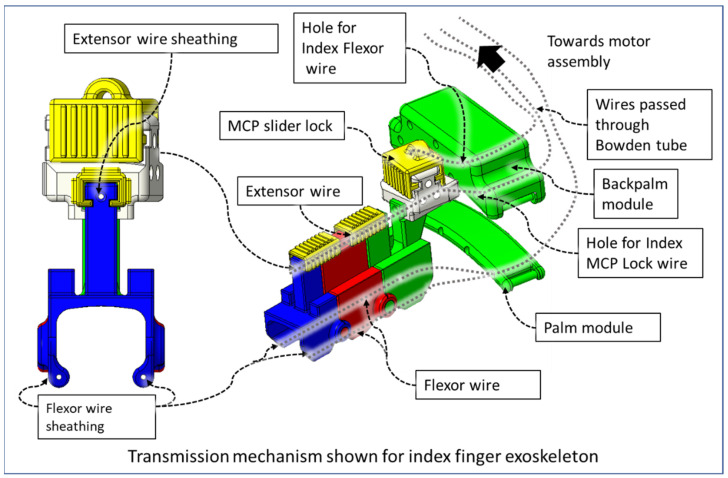
Cable-driven transmission mechanism of the device.

**Figure 9 micromachines-12-01274-f009:**
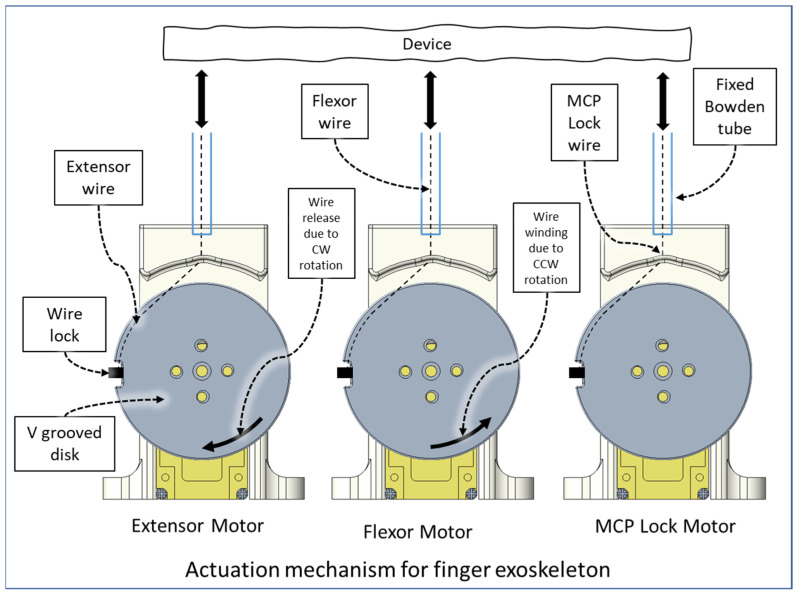
Actuation mechanism.

**Figure 10 micromachines-12-01274-f010:**
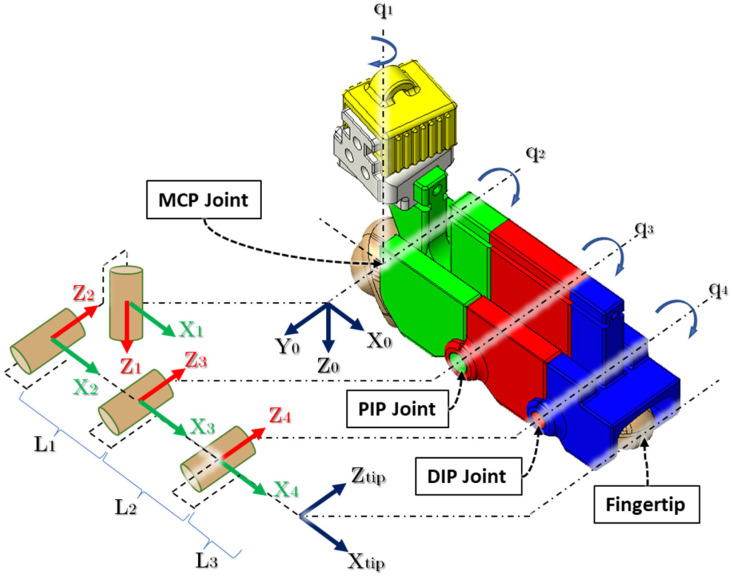
Kinematic chain of finger and exoskeleton.

**Figure 11 micromachines-12-01274-f011:**
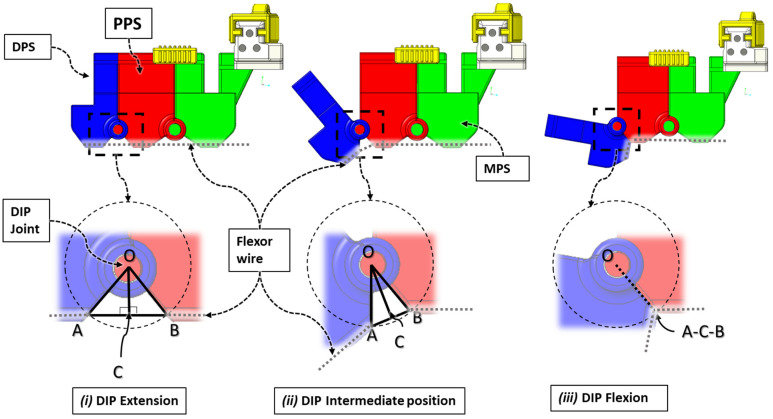
Relation between varying DIP joint flexion angles and effective flexion wire length.

**Figure 12 micromachines-12-01274-f012:**
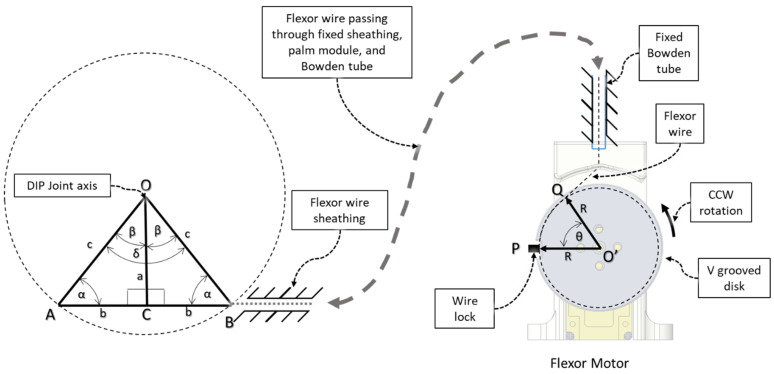
Relation between DIP flexion angle (left) and flexor motor’s angular rotation (right).

**Figure 13 micromachines-12-01274-f013:**
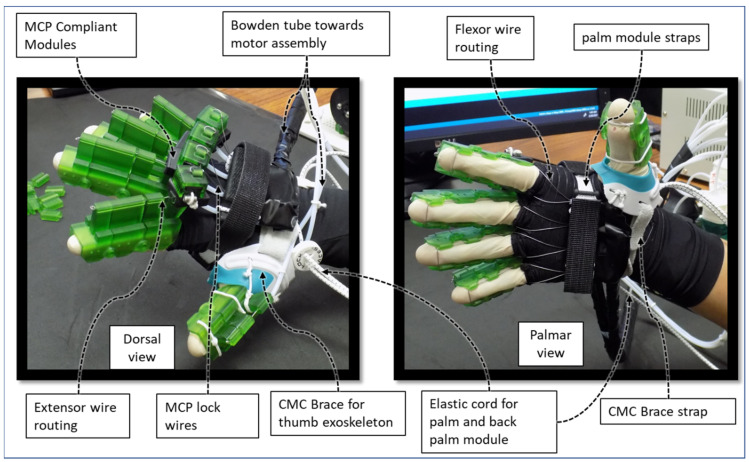
Palmar and dorsal view of the device worn by Participant-A.

**Figure 14 micromachines-12-01274-f014:**
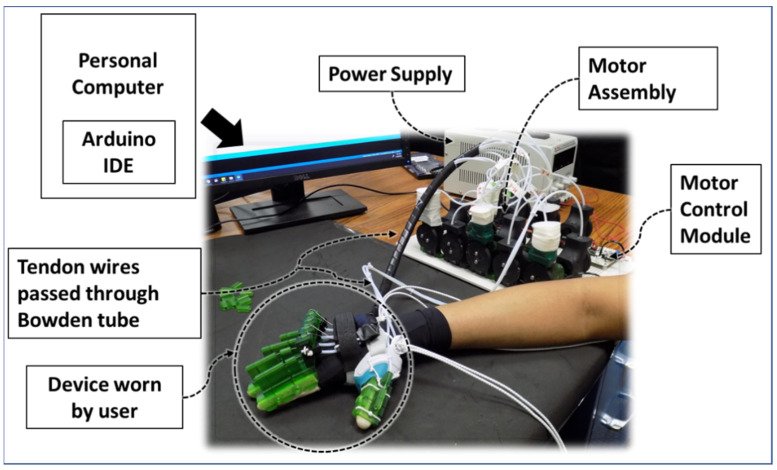
Schematic of the experimental setup.

**Figure 15 micromachines-12-01274-f015:**
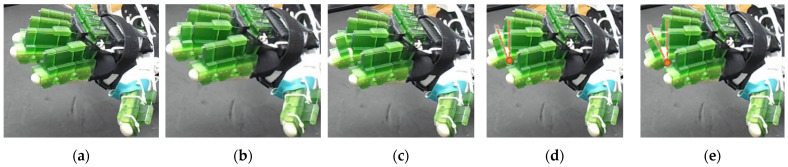
Isolated DIP flexion of middle finger: (**a**) hand is at extended position, DIP joint angle: 0°; (**b**–**d**) intermediate DIP joint positions; and (**e**) DIP joint angle: 35°.

**Figure 16 micromachines-12-01274-f016:**
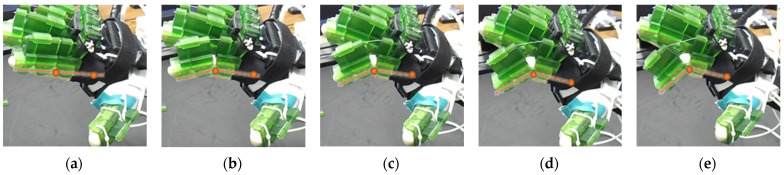
Isolated PIP flexion of index finger: (**a**) hand is at extended position, PIP joint angle: 0°; (**b**–**d**) intermediate PIP joint positions; and (**e**) PIP joint angle: 70°.

**Figure 17 micromachines-12-01274-f017:**
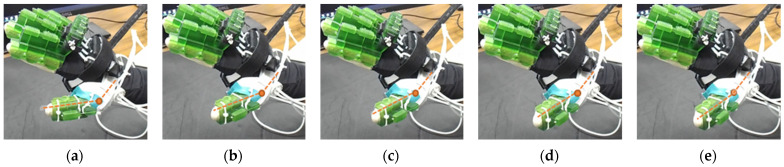
Isolated MCP extension of thumb: (**a**) thumb is at the flexed position, MCP joint angle: 25° (**b**–**d**) Intermediate MCP joint positions (**e**) thumb is at the extended position, MCP joint angle: 0°.

**Figure 18 micromachines-12-01274-f018:**
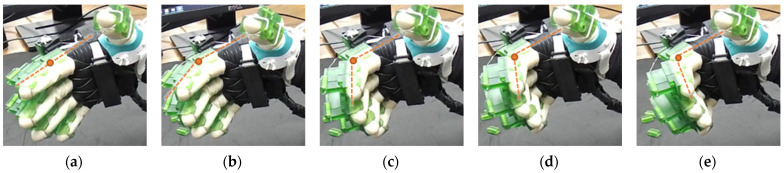
Fingers’ PIP flexion: (**a**) index, middle, ring, and small finger PIP joints are extended, PIP joint angle: 0°; (**b**–**d**) intermediate PIP joint positions; and (**e**) PIP joints are flexed, PIP joint angles: 80–82°.

**Figure 19 micromachines-12-01274-f019:**
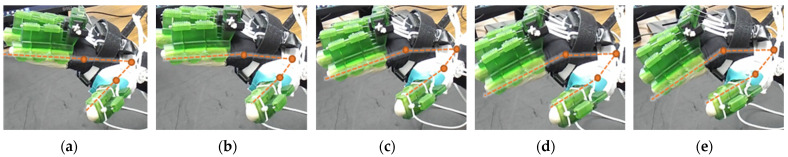
MCP flexion of fingers and thumb: (**a**) fingers and thumb are at the extended position, MCP joint angle: 0°; (**b**–**d**) intermediate MCP joint positions; and (**e**) fingers and thumb are at flexed, MCP joint angle, fingers: 55–57° and thumb: 25°.

**Figure 20 micromachines-12-01274-f020:**
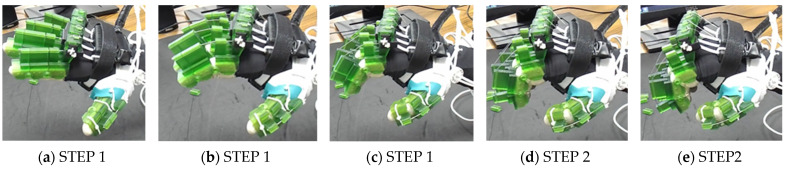
DIP, PIP, and MCP flexion of fingers and thumb: (**a**) all joints are extended, joint angles: 0°; (**b**–**d**) intermediate joint positions; and (**e**) all joints are at flexed.

**Table 1 micromachines-12-01274-t001:** Various hand exercises and associated digits and digit joint motions.

Hand Exercises	Associated Digits	Associated Digit Joint Motions
Knuckle bend	Index, middle, ring, and small finger	MCP F/E
Hook Fist		DIP-PIP F/E
Straight fist		PIP-MCP F/E
Imaginary ball squeeze/claw hand	Index, middle, ring, and small finger and thumb	DIP-PIP-MCP (fingers), IP-MCP (thumb) F/E
Isolated DIP F/E	Index/middle/ring/small	DIP F/E
Isolated PIP F/E	Index/middle/ring/small/thumb	PIP (fingers), IP (thumb) F/E
Composite F/E	Isolated or combination of index, middle, ring, small, and thumb	Combination of DIP, PIP, MCP (fingers); IP, MCP (thumb) F/E
F/E: flexion/extension; MCP: metacarpophalangeal joint; DIP: distal interphalangeal joint; PIP: proximal interphalangeal joint; IP: interphalangeal joint.		

**Table 2 micromachines-12-01274-t002:** Isolated and composite finger joint flexion–extension exercises.

Finger Joint Motions	Configuration Description
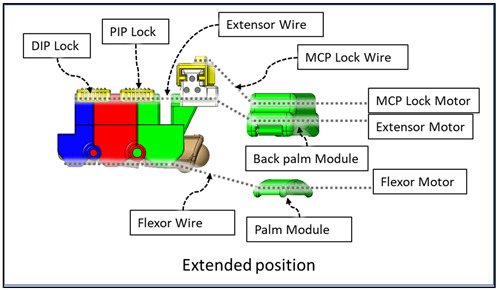	The figure shown in the right column shows the nominal/zero position of the device while the user is wearing it. The same configuration is achieved during extension exercises as such: the extensor motor pulls the extensor wire at a slower rate while the MCP lock motor (if moved from zero position) and flexor motors release associated wires. The DIP and PIP locks can be slid off from the exoskeletal shells.
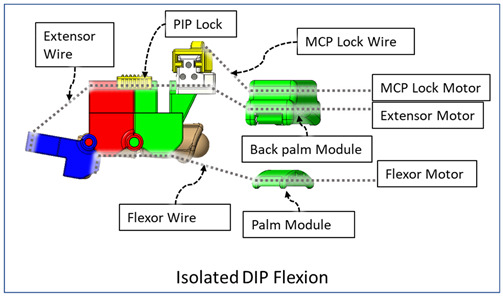	The DIP lock has been removed while the PIP lock stays. The MCP lock motor stays at zero position throughout DIP flexion to prevent MCP joint motion. The extensor motor releases the extensor wire at a faster rate while the flexor motor pulls the flexor wire.
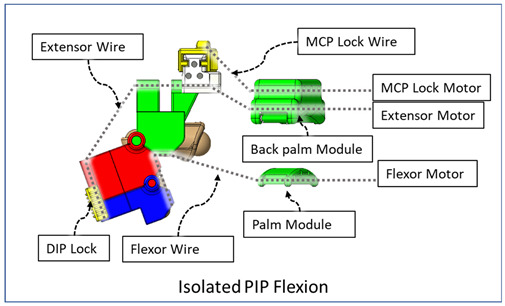	The DIP lock has been removed; the PIP lock has been slid to the DIP lock’s position and used as the DIP lock. The motor configuration is the same as that of the isolated DIP flexion.
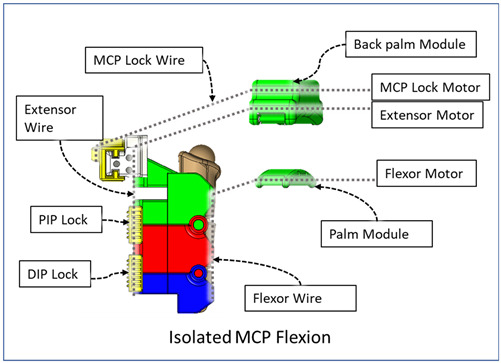	Both DIP and PIP locks stay at the locking position. During this motion, the extensor and MCP lock motors release the extensor and MCP lock wire, respectively, faster than the flexor motor pulls the flexor wire.
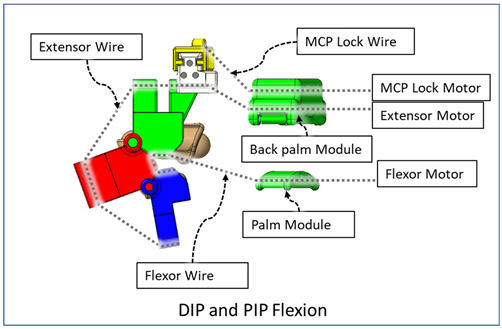	During this composite joint motion, both the DIP and PIP locks are slid off. During this motion, motor configurations are the same as the that of the isolated MCP flexion exercise.
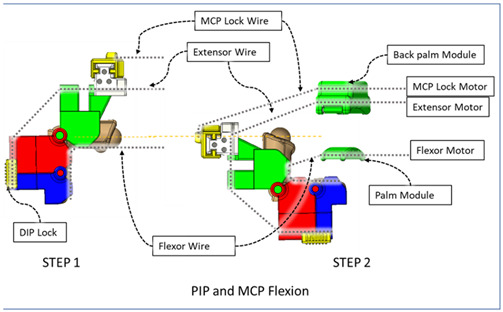	One sliding lock is kept to lock the DIP joint’s motion and is thus named as the DIP lock. At Step 1, PIP flexion is achieved during this composite joint motion through the same motor configuration as that of the isolated PIP flexion. At Step 2, the MCP lock wire is released simultaneously so that MCP flexion is performed.
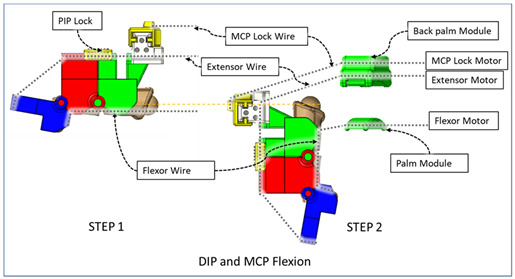	The PIP lock is kept at the joint locking position. At Step 1, DIP flexion is achieved during this composite joint motion through the same motor configuration as that of the isolated DIP flexion. At Step 2, the MCP lock wire is released simultaneously so that MCP flexion is achieved.
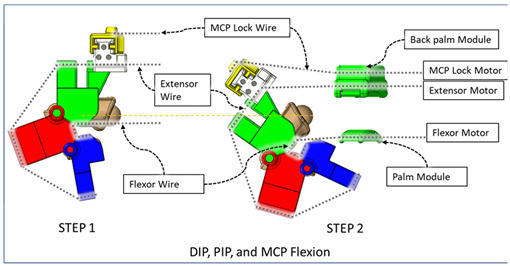	At Step 1, DIP and PIP flexion is achieved by keeping the MCP lock motor at zero position until the extensor and flexor motors have achieved DIP and PIP flexion. At Step 2, the MCP lock wire is released simultaneously with the extensor motor to achieve MCP flexion.

**Table 3 micromachines-12-01274-t003:** Modified DH parameters.

Link.	*α* * _i_ * _−1_	a*_i_*_−1_	*d* * _i_ * _−1_	*θ* * _i_ *	Joint Axis Associated Motions
1	0	0	0	q_1_	MCP abd/add
2	*π*/2	0	0	q_2_	MCP flex/ext
3	0	*L* _1_	0	q_3_	PIP flex/ext
4	0	*L* _2_	0	q_4_	DIP flex/ext
Fingertip (f)	0	*L* _3_	0	0	-

**Table 4 micromachines-12-01274-t004:** Calibration chart for finger joint angles.

**(i) DIP Flexion–Extension of Finger**				**(ii) PIP Flexion–Extension of Finger**			
Flexor motor’s angular position *θ*_F_ (°)	Extensor motor’s angular position *θ*_E_ (°)	MCP lock motor’s angular position *θ*_ML_ (°)	DIP joint angle (approx.) *θ*_d_ (°)	Flexor motor’s angular position *θ*_F_ (°)	Extensor motor’s angular position *θ*_E_ (°)	MCP lock motor’s angular position *θ*_ML_ (°)	PIP joint angle (approx.) *θ*_p_ (°)
51	0	0	60	80	0	0	90
48	7	0	55	75	6	0	85
44	12	0	52.5	70	12	0	80
41	16	0	50	65	18	0	75
38	21	0	45	60	24	0	70
34	26	0	40	55	30	0	62.5
31	28	0	35	50	37	0	47.5
28	33	0	32.5	45	43	0	42
25	35	0	25	40	49	0	37.5
21	40	0	20	35	55	0	32.5
18	45	0	17	30	61	0	27.5
15	50	0	14	25	67	0	22.5
11	54	0	10	20	73	0	17.5
8	57	0	8	15	79	0	12.5
5	61	0	5	10	85	0	7.5
3	64	0	2.5	5	91	0	2.5
0	67	0	0	0	97	0	0
**(iii) DIP and PIP Flexion-Extension of Finger**				**(iv) MCP Flexion-Extension of Finger**			
Flexor motor’s angular position *θ*_F_ (°)	Extensor motor’s angular position *θ*_E_ (°)	MCP lock motor’s angular position *θ*_ML_ (°)	DIP + PIP joint angle (approx.) *θ*_dp_ (°)	Flexor motor’s angular position *θ*_F_ (°)	Extensor motor’s angular position *θ*_E_ (°)	MCP lock motor’s angular position *θ*_ML_ (°)	MCP joint angle (approx.) *θ*_m_ (°)
98	13	0	95	110	0	0	65
91	24	0	90	103	4	9	60
85	35	0	85	95	7	13	58
78	45	0	79	88	9	18	55
71	56	0	70	81	11	22	50
65	65	0	56	73	13	26	40
58	75	0	50	66	15	31	33
52	84	0	43	59	18	35	30
45	95	0	36	51	20	40	25
38	105	0	31	44	22	44	20
32	117	0	25	37	24	48	18
25	127	0	19	29	26	53	16
19	137	0	14	22	29	57	12
12	146	0	9	15	31	62	8
7	156	0	3	7	33	66	4
0	164	0	0	0	35	70	0

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
