# Peer review of "Flexohand: A Hybrid Exoskeleton-Based Novel Hand Rehabilitation Device"

_micromachines, 2021, doi:10.3390/mi12111274_

Round 1
Reviewer 1 Report
This article presents the project of a hybrid Exoskeleton based Novel Hand Rehabilitation Device named Flexohand. It is interesting, but the Authors should improve the description of the adjusting and validation (calibration) process of this exoskeleton. The authors didn't explain is possible to use it by a whole group of people and how is it adjusting to each person?
Or, is it made for an individual?
Also, It is very important to fit the size of the exoskeleton to the hand's size. The authors didn't check do joints of each component of the exoskeleton have a rotation pivot at the same joints of the fingers?
The authors also didn't give information on how many actuators the prototype of Flexohand includes, and how is prepared a program of therapies? Is it prepared/obtained by the doctor or user?
Finally, the authors didn't contend: is calibration only need to do before first use or before each use?
Additionally, I don't agree with the authors' opinion that existing devices cannot allow wrist and forearm movement during finger exercises, maybe the authors would want to say that existing devices cannot control wrist and forearm movement during finger exercises.
I noticed some mistakes in the text as:
- please explain the abbreviation "MCP" in the first use at abstract,
- in Figure 1 please mark (A or B) which is the structure of the finger and which is the structure of the thumb,
- in line 381 is a mistake; it is OA = distance between O and B, instead: OA = distance between O and A.
I recommend publishing this article after minor revision (corrections to minor methodological errors and text editing to above-presented comments).
Reviewer 2 Report
Given the importance of home-based hand rehabilitation, the designed device shows good promise in terms of applicability. One of the main advantages of the device is its lightweight and the ability to permit wrist movement during finger exercise.
I suggest that the “Introduction” be reduced in size. Also, some of the kinematic equations can be put in the appendix. For example, some of the analyses that are related to equations (3.2) to (3.6) had better be moved to the appendix. On the whole, it is good research work.
